# Exploring perceived barriers, drivers, impacts and the need for evaluation of public involvement in health and social care research: a modified Delphi study

D Snape,[1] J Kirkham,[2] N Britten,[3] K Froggatt,[4] F Gradinger,[3] F Lobban,[4] Jennie Popay,[4] K Wyatt,[3] Ann Jacoby[1]

▶ Prepublication history and additional material is available. To view please visit the journal (http://dx.doi.org/10.1136/bmjopen-2014-004943).

For numbered affiliations see end of article.

**Correspondence to**
Professor Ann Jacoby;
ajacoby@liv.ac.uk

## ABSTRACT

**Objective:** To explore areas of consensus and conflict in relation to perceived public involvement (PI) barriers and drivers, perceived impacts of PI and ways of evaluating PI approaches in health and social care research.

**Background:** Internationally and within the UK the recognition of potential benefits of PI in health and social care research is gathering momentum and PI is increasingly identified by organisations as a prerequisite for funding. However, there is relatively little examination of the impacts of PI and how those impacts might be measured.

**Design:** Mixed method, three-phase, modified Delphi technique, conducted as part of a larger MRC multiphase project.

**Sample:** Clinical and non-clinical academics, members of the public, research managers, commissioners and funders.

**Findings:** This study found high levels of consensus about the most important barriers and drivers to PI. There was acknowledgement that tokenism was common in relation to PI; and strong support for the view that demonstrating the impacts and value of PI was made more difficult by tokenistic practice. PI was seen as having intrinsic value; nonetheless, there was clear support for the importance of evaluating its impact. Research team cohesion and appropriate resources were considered essential to effective PI implementation. Panellists agreed that PI can be challenging, but can be facilitated by clear guidance, together with models of good practice and measurable standards.

**Conclusions:** This study is the first to present empirical evidence of the opinions voiced by key stakeholders on areas of consensus and conflict in relation to perceived PI barriers and drivers, perceived impacts of PI and the need to evaluate PI. As such it further contributes to debate around best practice in PI, the potential for tokenism and how best to evaluate the impacts of PI. These findings have been used in the development of the Public Involvement Impact Assessment Framework (PiiAF), an online resource which offers guidance to researchers and members of the public involved in the PI process.

## Strengths and limitations of this study

- Despite growing interest in the potential benefits of public involvement (PI) in health and social care research, there has been little examination of how different stakeholders perceive the barriers, drivers, impacts and need for evaluation. As part of a larger study to develop guidance on assessing PI impacts, we undertook a mixed-method modified Delphi study which has provided primary evidence of areas of consensus and conflict around these issues.
- This study involved a heterogeneous panel of PI experts, reflective of the range of key stakeholder groups and was geographically diverse; 'consensus' thresholds were determined in advance of data collection.
- A limitation of the study was that response rates were relatively low, so that our conclusions are potentially biased. However, study reliability and validity were enhanced by providing panellists with the opportunity to comment on their views and on the views of others via open feedback; and the quality of the data obtained was high.
- This study is the first, to our knowledge, to present empirical evidence of the opinions of key stakeholders about the impacts of PI; and to identify areas of consensus and conflict around these impacts.
- We have also identified a number of key issues in relation to perceived PI barriers and drivers and approaches to the evaluation of PI in health and social care research. In particular, our respondents have highlighted that tokenism around PI represents a 'self-fulfilling prophecy', best addressed through development of clear guidance and measurable standards.

## INTRODUCTION

Internationally[1] and within the UK[2 3] interest in the potential benefits of public involvement (PI) in health and social research has grown; and in parallel, there has been increasing

demand for researchers to articulate and demonstrate the value of PI to funding bodies.[4]

While a considerable body of literature about PI in research reports on the process of involvement,[5–8] such accounts often fall short in their description of precisely what differences PI made to the research process and/or outcomes.[9] There has been relatively little high-quality research effort around assessing the impact of PI[9–14] possible reasons being that evaluation is too difficult and that PI is of intrinsic value and as such needs no further justification.[9 15–17] Conversely, other authors have articulated counterarguments for evaluating impact, which broadly relate to the issues of effectiveness, ethics, economics and the need for evidence.[13 14 18 19] Within the health research community, opinion about the value of PI appears divided with some researchers arguing that it represents a threat to the quality or robustness of research design[20 21] and data collection[22 23] and others proactively embracing the PI endeavour.[15–17 24] We would argue that evidence of the impacts of PI is important for a number of reasons: first, to ensure research integrity; second, to maximise PI impact and so improve research quality; third, to minimise the possibility of any negative effects on the research and on those involved and last, to justify the use of research resources to support PI.

The aims, objectives and methods of the modified Delphi study reported here have previously been described in detail elsewhere.[25] The Delphi study was part of a larger study that aimed to produce a Public Involvement Assessment Framework and related guidance (PiiAF; see piiaf.org.uk). In the present paper, we focus our exploration on areas of consensus and conflict around barriers and drivers to PI in research, perceived impacts of PI and whether and how these should be evaluated.

## METHODS
### Delphi technique
Originally developed by the Research and Development (RAND) Corporation for technological forecasting, the Delphi technique has been used extensively within health and social science research.[26–31] The technique rests on the assumption that group opinion carries greater validity than individual opinion; and as such, it offers a reliable data collection method to explore the opinions of a group and seek to identify consensus in circumstances where there is uncertainty or paucity of knowledge surrounding the topic area under investigation.[32–35] Since its inception, subsequent users of the Delphi technique have modified its process and no universal Delphi design is dominant.[33 34 36] Similarly, variations in panel size[37] as well as numerous variations in the criteria for judging consensus agreement between participants[34 35 38 39] have been reported. The Delphi technique has also been criticised, as it is perceived to

force consensus and to be weakened by not allowing panellists to elaborate on their views.[27]

For this reason the current Delphi study used a modified technique where consensus was not sought; rather panellists were provided with opportunities to elaborate on why they held the views they expressed or endorsed[28] and an attempt was made to tease out areas of conflict as well as areas of consensus.

Despite variations in approach, there are a number of characteristics which, in combination, distinguish the basic Delphi technique from other research methods. These are anonymity, multistage iteration and controlled feedback, exploration of consensus via statistical group response and the use of a panel of experts.[35 40] Each of these characteristics was given due consideration in the present study, in order to enhance the validity and reliability of the research design and the quality of responses.[33 41 42]

### Modified Delphi process and 'expert' sample
Details relating to the mixed-methods approach used were previously reported by Snape et al.[25] However, in brief, (see table 1) the modified Delphi study from which these data are drawn was conducted between November 2011 and September 2012, and consisted of the following three stages:

▶ Three 'expert' workshops (participant total n=42) including members of the public, academic, clinical and user-researchers, research funders and research managers that explored issues around values and debates underpinning PI in order to develop questions for rounds 1 and 2 of the modified Delphi survey.

▶ A pilot study involving 11 participants (academics, n=6; user-researchers, n=3; Patient Advisory Group members, n=2), to test the round 1 survey questionnaire, in which careful attention was paid to the content and layout of the invitation email, the survey layout and the clarity of questions. Language, question type and questionnaire formatting were edited in response to participant feedback.

▶ An online, two-round, modified Delphi survey (231 panellists participated in both survey rounds) to explore areas of consensus and conflict around the values underpinning PI and the barriers and drivers, perceived impacts in health and social care research and ideas about how to assess these impacts. Questions relating to the issues that are the focus of this paper are reproduced in online supplementary appendix 1. Where an issue considered at round 1 was felt to require further exploration it subsequently was pursued in round 2.

For the purposes of the Delphi process we defined PI as an active partnership between members of the public and researchers in the research process, rather than the use of people as the 'subjects' of research; and used the definition of 'public' offered by the UK National

**Table 1** The modified Delphi process

| Criteria | Expert workshops | Pilot testing | Round 1 survey | Round 2 survey |
|---|---|---|---|---|
| Panel size | *Northwest*<br>Invited n=25<br>Attended n=15<br>*Southwest*<br>Invited n=25<br>Attended n=19<br>*Public advisory group*<br>Invited n=11<br>Attended n=8 | Invited n=11<br>Responded n=10 | Invited n=740<br>Opted-out n=23<br>Responded at round 1 n=318 | Eligible n=318<br>Opted-out of round 2 n=3<br>Invited to participate in round 2 n=315<br>Responded at round 2 n=231 |
| Reminders | NA | Yes×1 | Yes×2 | Yes×2 |
| Response rate | NA | 91% | 43% | 73% (of 43%) |
| Area of expertise | Members of the public<br>User/academic/clinical researchers<br>Research managers<br>Research commissioners | Members of the public<br>User/academic/clinical researchers<br>Research managers<br>Research commissioners | Members of the public<br>User/academic/clinical researchers<br>Research managers<br>Research commissioners | Members of the public<br>User/academic/clinical researchers<br>Research managers<br>Research commissioners |
| Problem exploration | Round-table discussions/group activities to explore normative debates around the value/potential impacts of PI | Questionnaire—questions derived from literature review and Expert Workshop outcomes with five-point and seven-point Likert scales for close-ended questions<br>Open question options | Questionnaire—as for pilot testing with revisions to unclear questions and formatting<br>Additional open questions added to provide further opportunities for comment | Questionnaire—questions derived from analysis of round 1 responses with five-point Likert scale for close-ended questions |
| Consensus | NA | NA | 70% endorsement with at least 55% in the extreme category=*critical* consensus<br>60% endorsement=*clear* consensus | 70% endorsement with at least 55% in the extreme category=*critical* consensus<br>60% endorsement=*clear* consensus |
| Feedback | Expert Workshop outcomes fed back to participants and members of the Public Advisory Group | Consultation process | Expert panel members fed back responses with response %age of their own sub-group and those of other sub-groups<br>Summaries of comments made by respondents also fed back | Wide-spread project dissemination of findings:<br>Study report(s)<br>Workshops<br>Conference presentation(s); peer-reviewed journal publication(s) |
| Access route(s) to data collection | Email<br>Group discussions<br>Video-conference | Email<br>Face-to-face<br>Tele-conference | Email<br>Online questionnaire | Email<br>Online questionnaire |

Advisory Group, INVOLVE,[43] where the term includes patients and potential patients, carers and people who use health and social care services.

The sampling strategy for panel selection was purposive across a number of 'expert' stakeholder groups.[44] 'Experts' were defined as a group of *informed individuals*[33] or those with knowledge or experience of a specific subject.[45 46] This approach enabled the recruitment of a large heterogeneous panel from which we aimed to capture diverse perspectives and interests around PI in research. Potential panellists were identified in one of three ways:

▶ Directly, through research team members' contacts and networks.
▶ Through conducting online searches of open-access research information and funding sites.
▶ Via a review of literature in the field of PI in health and social care research.

### Anonymity

Anonymity between panellists was guaranteed. At round 2 of the modified Delphi survey we fed back to panellists their own reactions to opinions and key arguments as well as levels of consensus for each of the subgroups. Each opinion carried the same weight and was afforded the same degree of importance in the analysis. In this way, subject bias was eliminated.[28] This approach enabled panellists to be open and honest about their views on various issues and to express an opinion without feeling pressured into conforming to the views of others.[28]

### Quantitative data analysis

As previously stated, published Delphi studies indicate there is no fixed level of consensus to employ.[34 35 38 39 47] Based on review of levels of consensus defined in other Delphi studies, the criteria for consensus (see table 2) were defined prior to data collection, to ensure statistical integrity, as follows:

▶ *Critical* consensus which represented 70% endorsement or rejection of a statement, with at least 55% of responses endorsed or rejected using the extreme categories (ie, strongly agree, strongly disagree).

▶ *Clear* consensus which represented 60% endorsement or rejection of a statement. Where responses clustered in one response option only, consensus was not assumed and this item was further explored in round 2 of the survey. Also explored in round 2 were 'unexpected' (as defined by the study team) endorsements of items by the subgroups.[25]

### Qualitative data analysis

Qualitative analysis of responses in the text boxes at round 1 and round 2 allowed further exploration of quantitative findings. Thematic codes were identified using framework analysis.[48] The data were analysed by DS. Coding, categorisation and quality checking were conducted collaboratively with AJ, who also reviewed 10% of the qualitative data. Data were first reviewed inductively to identify recurring themes and concepts raised by participants; these were coded and formed the initial major and subthemes. Additional codes were then incorporated through an iterative process involving DS and AJ. The thematic framework was further refined before being applied systematically to the whole data set. This process facilitated identification of any inconsistencies in coding, which were subsequently discussed and reconciled.

### PI in the Delphi study

The public was involved in the Delphi study in a number of ways: as service-user researchers on the main PiiAF study team; as members of the PiiAF project's Public Advisory Group (PAG) and of the National Advisory Network.

Members of the PAG contributed to all phases of the modified Delphi study. Specifically, at the first expert workshop PAG members were able to debate and consider values around PI in health service research, including value consensus and conflicts; value rankings and impacts; value statements and their categorisation and how conflicts might be accommodated in research policy and practice. At the second workshop members of the PAG were able to contribute to normative debates

| Table 2 | Examples of consensus definitions | | | | | |
|---|---|---|---|---|---|---|
| **Example statements** | **Agree strongly** | **Agree somewhat** | **Neither agree or disagree** | **Disagree somewhat** | **Disagree strongly** | **Total** |
| Statement 1 Public involvement can make a major difference to the way research findings are used to bring about change in service provision | 144 (48%) | 120 (40%) | 26 (9%) | 10 (3%) | 1 (<1%) | 301 |
| Statement 2 The public should be actively involved in any publicly funded research which may impact on their health status | 186 (62%) | 70 (23%) | 24 (8%) | 18 (6%) | 3 (1%) | 301 |
| Statement 1=clear consensus (sum of positive responses 60%+). Statement 2=critical consensus (sum of positive responses 70%+, with 55% saying, 'strongly agree'). | | | | | | |

**Table 3** Response percentage per stakeholder group at survey round 1 and round 2

| Stakeholder group | Round 1 n=318* response percentage per stakeholder group | Round 2 n=231 response percentage per stakeholder group |
|---|---|---|
| CA | 63 (20%) | 40 (17%) |
| NCA | 88 (28%) | 67 (29%) |
| MP | 55 (17%) | 41 (18%) |
| RM or funding/commissioning body employee | 76 (24%) | 56 (24%) |
| Occupying MR | 34 (11%) | 27 (12%) |

*Information about stakeholder group was missing for two panellists.
CA, clinical academic; MP, member of the public; MR, multiple roles; NCA, non-clinical academic; RM, research manager.

around PI in health service research; consider the roles of service users in carrying out varying kinds of research and identify PI tensions and reconciliation. PAG members participating in the third workshop were able to consider how the findings from workshops 1 and 2 might be translated into questions for rounds 1 and 2 of the modified Delphi survey; make suggestions for additional questions and/or further exploration of PI concepts and identify potential recruitment mechanisms for the modified Delphi survey sample.

We also had assistance with piloting the round 1 survey from PAG members, who suggested some changes in relation to a number of items. These included, for example, changes to: the content and wording of the survey participant introductory email, to ensure understandability and a 'user-friendly' format; the instructions/explanations provided in the survey documents to improve accessibility; the survey questions to improve their relevance and appropriateness, including the identification of potentially problematic questions. They also offered advice in relation to the ease of access and user friendliness of the format of the online survey programme; and on the potential acceptability of the time required to complete the online survey.

Members of the PAG were also involved in reviewing Delphi study reports and papers for publication in peer-reviewed journals and producing lay summaries.

## RESULTS
Panellists' perceptions of barriers and drivers to PI in research, of the potential impacts and of ways of assessing these were explored in both rounds of the survey. As in our earlier paper focusing on values around PI,[25] we therefore discuss the relevant findings from each round together.

### Delphi panellists
#### Survey round 1
Seven hundred and forty (n=740) potential 'expert' Delphi panellists were invited via email, to participate in the online survey. Up to two reminder letters were emailed, yielding a total response of 318 (RR=43%). Responding panellists self-selected themselves into one of five 'stakeholder' groups, as outlined in table 3.

High levels of expertise were reported by panellists (table 4), but despite high levels of expertise, fewer than half (n=134; 48%) had undergone formal training relevant to PI in health and social care research.

#### Survey round 2
Those panellists (n=318; RR=43%) submitting a questionnaire at round 1 were subsequently invited to participate in the round 2 survey. Of the 318 responders, three electronically 'opted out' of receiving further communication; therefore, the round 2 questionnaire was sent out to 315 panellists (table 3). As with round 1, two reminders were emailed to non-responders and a total of 231 responses were received, (response rate of 73% (of 43%)).

### Key factors that influence effective PI
At round 1, panellists were asked to consider a number of factors (as outlined in the online supplementary

**Table 4** Research experience by stakeholder group*

| Stakeholder group | Minimum 5 years research experience | Some PI responsibility | Formal training relevant to PI |
|---|---|---|---|
| CA | 52 (82.5%) | 52 (82.5%) | 27 (42.9%) |
| NCA | 70 (79.5%) | 63 (71.6%) | 27 (30.7%) |
| MP | 33 (60%) | 27 (49.1%) | 35 (63.6%) |
| RM or funding/commissioning body employee | 53 (69.7%) | 64 (84.2%) | 31 (40.8%) |
| Occupying MR | 30 (88.2%) | 29 (85.3%) | 14 (41.2%) |

*Data taken from round 1.
CA, clinical academic; MP, member of the public; MR, multiple roles; NCA, non-clinical academic; PI, public involvement; RM, research manager.

appendix 1) that likely impact either as a barrier or a driver to effective PI. The 21 factors were identified from data collected at our previously conducted workshops or from the extant PI literature; and related to the nature (12 items) and the interpersonal aspects (9 items) of the research process. On a seven-point scale from 'major barrier' through to 'major driver' panellists were asked to rate each item as either a barrier or a driver.

At round 1, there was critical consensus across all panellists for three, and clear consensus for one, major or moderate barriers to effective PI.

▶ Attitudes of researchers to relinquishing power and control (71% agreement);
▶ Scientific language used in research (70% agreement);
▶ Lack of support for PI from research funders (70% agreement);
▶ The perception that members of the public have biased views (63% agreement).

There was also clear consensus at round 1 around five major or moderate drivers to effective PI:

▶ The recognition that members of the public have a valuable contribution to make (69% agreement);
▶ Clear communication between research team members (67% agreement);
▶ Designated funding for PI (66% agreement);
▶ Time to build partnerships and trust (65% agreement);
▶ Training for researchers about PI (63% agreement).

At round 2, the 12 possible barriers or drivers for which there was no consensus at round 1 were presented back to panellists, who were asked to rank in order of importance which they regarded as the three greatest barriers and, similarly, the three greatest drivers. Three factors emerged as the most important barriers, the first two in the list being cited consistently and endorsed across all stakeholder groups:

▶ The attitudes of academic researchers/clinicians to involving the public in research;
▶ Perceived importance of PI;
▶ Lack of research experience of members of the public.

The three factors emerging as the most important drivers are identified below. Once again, the first two drivers in the list were cited consistently and endorsed across all stakeholder groups:

▶ Ability to be open and flexible to difference;
▶ Attitude of researchers;
▶ Perceived importance of PI in health and social care research.

Overall, at round 2 panellists recognised that the same factor when managed well could operate as a driver of PI while when managed poorly operated as a barrier. As one non-clinical academic explained:

> There are no major barriers if you want to do it… it is a lack of commitment and or interest in doing the necessary learning to do it well. When people do it badly it then reinforces their belief it is not of value. [NCA, round 2]

Open question responses highlighted that tensions across different stakeholder groups within health and social care research were seen as an inevitable consequence of collaborative working. Time to develop team cohesion as well as PI training for members of the public and researchers were seen as pivotal factors in effecting meaningful PI:

> There needs to be a recognition that all sides have valuable contributions to make to research and that peoples' attitudes and beliefs, both researchers and the public, are valid and worthy of respect. Training is important and draws the public into the team [NCA, round 2]

Panellists at both rounds repeatedly acknowledged that stakeholder motivation and the positive attitude of all involved were essential prerequisites for good PI. As one clinical academic explained:

> I was involved in a collaborative group that met consistently since 2007. It has been a journey of experience. Over time that understanding has evolved and grown about good public involvement. This experiential learning took theoretical ideas and made them a reality. It gave the opportunity to challenge the internal subtle prejudice that most clinicians have to public involvement to create a real working relationship that can produce research. [CA, round 1]

### Issues related to the potential for PI tokenism

Some panellists were of the opinion that tokenism in PI was value driven:

> The issue is a cultural one. In my experience, there are very, very few researchers, scientists, doctors who really value public input and involvement. It is done because it ticks the boxes for funding, but the attitude is of resigned tolerance rather than a view that the public add value. [MP, round 1]

On a more positive note it was argued by one research manager that:

> Changing cultures takes time and three years into my role, I am starting to see results. [RM, round 2]

It was felt that PI needed to be embedded into the culture of organisations; not least by challenging those whose PI endeavour was suggestive of tokenistic practice. Perspectives on potential barriers and drivers to PI were further explored at round 2 when panellists were asked to suggest what, in their opinion, needed to change in order to make PI more than just 'tokenistic'. A number of key themes emerged from the data. These included:

▶ The need to provide clear guidance on the purposes of PI, together with models of good practice and measurable standards;
▶ The provision of and access to appropriate PI education and support for members of the public and clinical and non-clinical academic researchers;

- The need for hosting institutions, research ethics committees, journals and funders to be more proactive in facilitating and embedding PI within infrastructure systems and in promoting the reporting of PI;
- The need to redress the power imbalances in the research process which are felt to favour clinical and non-clinical academic researchers;
- The need for adequate resources, including the provision of funding early on (ie, preprotocol) to enable PI to be embedded early on in the research process.

Our data indicate that mediators to effective PI appeared to fit into two main categories: microlevel mediators including, for example, development of people skills, development and subsequent management of team dynamics; and macrolevel mediators including the quality of organisational infrastructures to support PI. Panellists suggested that training for members of the public should involve more than just an overview of research methods; it also needed to include education about political and policy context(s), as well as address any aspects of personal development training which people identified.

Our panellists also commented that effective PI is embedded in partnership and process values—doing good PI involves the development of relationships. This finding supports the position of INVOLVE[43] who promote active 'partnerships' with members of the public in the research process, emphasising the need for engagement, support and training. Interestingly, many panellists expressed the view that the process of involvement, when carried out well, is often difficult to deconstruct in order to evaluate discrete elements of the PI contribution and/or impact.

### Issues related to impacts of PI

At round 1, panellists were asked to consider 13 impact statements (see online supplementary appendix 1). There was consensus for 10 of the 13 statements, with critical consensus among panellists for three and clear consensus for seven of the statements (figure 1).

However, many panellists also commented that assessing how PI influences a research project is methodologically challenging, as articulated by the following two panellists:

At one level, it is about involving people in a positive way, ensuring their experience of research is constructive and meaningful. Effective implementation is also about the involvement meeting the goals or purpose intended, so that would need to be assessed against these, which are usually project-specific. Often, this will be looking at how the research is different as a result of public involvement, but sometimes that is difficult to discern and may not be very dramatic (if the research has been designed well in the first place). Also, public involvement may not result in changes to the research, but achieves greater acceptance of the research in the relevant communities and that may be difficult to assess. [RM, round 1]

Each research project is different and has different objectives for public involvement so it is hard to evaluate scientifically what the effects are. [DR, round 2]

Non-clinical academics were the group that most strongly endorsed the position that assessing how PI influenced research was methodologically challenging. Seventy-one per cent strongly agreed/agreed, compared with 56% of members of the other stakeholder groups. A somewhat surprising finding was that despite high endorsement of the potential positive impacts of PI in research, there was no consensus that it necessarily improves the quality and relevance of research. Members of the public were most likely to think (55%) that PI leads to research of greater quality and relevance; while academic researchers were least likely to think this (32%). Likewise, there was no consensus across the stakeholder groups for the statement that PI makes it more likely that findings from research will be used. However, as one clinical academic pointed out:

Absence of evidence isn't evidence of absence and just 'cos we can't yet demonstrate the impact of PI on research quality and relevance it doesn't mean we never will. As the body of evidence grows the likelihood of showing how and whether PI impacts on research quality and relevance grows and views on this may change [CA, round 2]

Given the level of agreement about methodological difficulties in assessing PI, we asked panellists at round 2 to consider how important they felt it was to do so. Overall, panellists expressed the view that assessment of PI was either very (58%) or fairly (31%) important, only

**Figure 1** What are the impacts of public involvement (PI) in health and social care research?

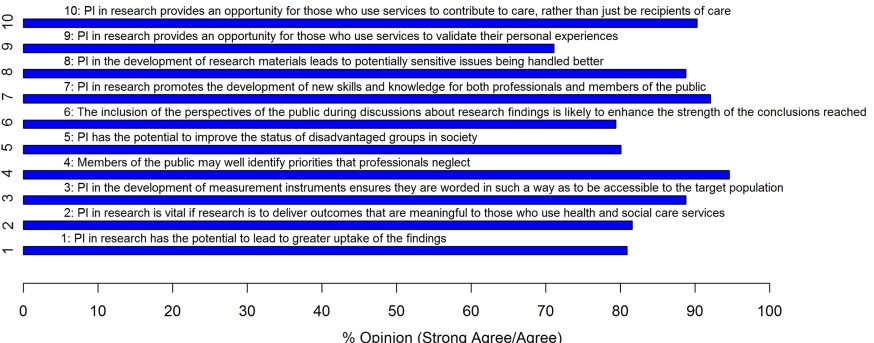

10: PI in research provides an opportunity for those who use services to contribute to care, rather than just be recipients of care
9: PI in research provides an opportunity for those who use services to validate their personal experiences
8: PI in the development of research materials leads to potentially sensitive issues being handled better
7: PI in research promotes the development of new skills and knowledge for both professionals and members of the public
6: The inclusion of the perspectives of the public during discussions about research findings is likely to enhance the strength of the conclusions reached
5: PI has the potential to improve the status of disadvantaged groups in society
4: Members of the public may well identify priorities that professionals neglect
3: PI in the development of measurement instruments ensures they are worded in such a way as to be accessible to the target population
2: PI in research is vital if research is to deliver outcomes that are meaningful to those who use health and social care services
1: PI in research has the potential to lead to greater uptake of the findings

% Opinion (Strong Agree/Agree)

a minority believing PI assessment to be unimportant. Across stakeholder groups, the proportion endorsing PI assessment as 'very important' ranged from 40% to 75%.

A number of panellists observed that to evaluate PI in isolation was 'discriminatory'; rather, it was argued, all aspects of the research process required evaluation. A number of justifications for undertaking PI evaluation were cited and included the suggestion that evaluation provides a mechanism for examining policy and practice in relation to PI, and can be an advocate for change. In the comment below a clinical academic describes how evaluation of PI within her own research team had led to changes in PI practice:

> We now put more thought and preparation in to what we want the public members to contribute from the outset. If they are involved in developing research questions then it is more likely that their participation will be meaningful at subsequent stages. For each study we now develop a job specification of what is expected, as the basis for discussion and when multiple public members want to participate, to guide selection. It has made the process more formal but it has forced us to think through how and when involvement would be meaningful study by study. [CA, round 2]

At round 1 there was no consensus among panellists about the contribution of PI to improving the quality and relevance of research, or the ways in which research is used. In response to these round 1 findings, panellists were asked, at round 2, to consider whether lack of agreement about the contribution of PI to improving these elements undermined its intrinsic value. Over half the panellists (58%, ranging from 42% to 67% across stakeholder groups) said they did not believe this to be the case, but that a number of issues likely contributed to this lack of agreement—a key challenge being the lack of a common understanding as to the what, when and how of PI. Panellists articulated that questions about the value of PI were answerable only by good evidence. However, lack of sophistication in identifying the unique contribution of PI to the research process, together with lack of clarity around its implementation and practice made meaningful evaluation problematic.

The fact that only 33% and 35% of clinical and non-clinical academic researchers, respectively, said PI *added value* to research was felt by some panellists to be 'damaging to the PI cause' and was perceived as 'a lever for providing academics with the excuse not to participate in future PI' Conversely, others argued that the *no value* perception put forward by the academic community should not be interpreted as *PI not having value* but rather as a reflection of the way in which academics themselves practiced PI—that is tokenistically:

> If it is not seen to have value it is less likely to be embedded and will thus remain tokenistic without reaching its full potential value. [NCA, round 2]

## DISCUSSION

Through an online, two-round modified Delphi survey involving a range of stakeholder groups we explored areas of consensus and conflict around perceived barriers and drivers to PI in research, perceived impacts of PI and possible approaches to its evaluation in health and social care research. The Delphi approach enabled data to be drawn from a large, geographically dispersed, heterogeneous panel of people with extensive experience of and expertise in PI in research across a range of stakeholder groups.[44] Panellists' responses were fairly evenly dispersed across the various stakeholder groups and the response rate of 43% was, in our view, acceptable.[49–51] The reliability of the study and the validity of the results were enhanced by providing panellists with the opportunity to comment on their views and on the views of others from the previous round via open feedback.[41]

### Key themes

There were high levels of consensus about the most important barriers and drivers to PI, although there were a number of other factors for which consensus was less clear. Perhaps inevitably, the most frequently endorsed drivers of PI were, in essence, the well-managed opposites of the most frequently endorsed barriers. In this respect, they can all be seen as factors which will likely influence, for better or worse, the impacts of PI. They therefore offer a useful checklist for research teams wishing to maximise the impact of PI. Our findings suggest that restrictions around research funding, funding mechanisms for paying people for their time and endeavour, together with existing workload time pressures were among some of the barriers to meaningful PI identified by many panellists. Staniszewska *et al*[13] identified similar process-related barriers associated with effective PI implementation which may go some way to explaining the disparities between current PI rhetoric and its practice.[52] Encouragingly, recent evidence suggests that even small-scale financial support for involving members of the public in research processes—in these examples at the grant development phase—can have positive impacts.[53 54] For example, Walker and Pandya-Wood[54] evaluated effectiveness of a prefunding bursary scheme and concluded that for a relatively small outlay appropriate involvement was possible, enabling refinement of the research question and design, encouraging team building and providing a useful learning opportunity for researchers and service users.

Team building endeavours, a positive attitude towards PI and the ability of research team members to be open and flexible to the perspectives of others were seen to be necessary prerequisites for facilitating effective PI. The majority of panellists across all stakeholder groups articulated the importance of appropriate training for researchers and members of the public, which would facilitate positive engagement and a shared

understanding of team members' roles. Panellists identified advice and mentoring schemes and financial reimbursement for public/service users involved in research as possible ways of supporting team cohesion. This finding is echoed by NIHR Research Design Service strategy and provision[55]; and an NIHR-wide 'Learning for Involvement' working group established and supported by INVOLVE will shortly report on the key messages from their consideration of how training and development for PI in research should be supported.

There were high levels of consensus across 10 impact statements. However, despite much positive endorsement of the potential benefits of PI in research, there was no consensus that PI necessarily improves research quality and relevance. While there was support for the position that assessing PI impacts is methodologically challenging, there were high levels of consensus about the need to assess impact. Although PI was perceived by many panellists as having intrinsic value, the majority believed its intrinsic value did not and should not diminish the importance of evaluating its impact alongside other research processes and outcomes. However, there was also a strong belief that articulating and demonstrating the value of PI was made more difficult by tokenistic practice, since the impact of PI is highly dependent on the quality of its conduct and on the openness and clarity with which it is reported. We would argue therefore that PI tokenism presents itself as a self-fulfilling prophecy (figure 2): PI when undervalued leads to tokenism in involvement practice; tokenistic practice fails to demonstrate the value of PI; hence, PI is therefore perceived as not adding value to health and social care research. This attitudinal underpinning of tokenism may be further compounded by practical constraints and barriers as highlighted earlier in the paper. Thus, addressing tokenistic practice and any accompanying constraints and barriers robustly remains a priority for all stakeholders in the PI enterprise.

## Delphi study limitations

In this investigation, we opted to use a modified Delphi approach for data collection, with fixed choice and open questions, in order to try to maximise our understanding of the issues under consideration. Our survey approach places inevitable limits on the depth of the data obtained and it would be important to follow-up key issues using more in-depth approaches, thus facilitating more detailed exploration of less well understood and articulated issues.

McKenna[33] reported that face-to-face contact with participants at round 1 was a useful strategy for increasing the response rate in Delphi studies. However, due to the size of our sample, many of the panellists were targeted 'cold', without prior notice. This approach may have had an impact on our round 1 response rate. In light of this, two reminder cover letters were emailed to

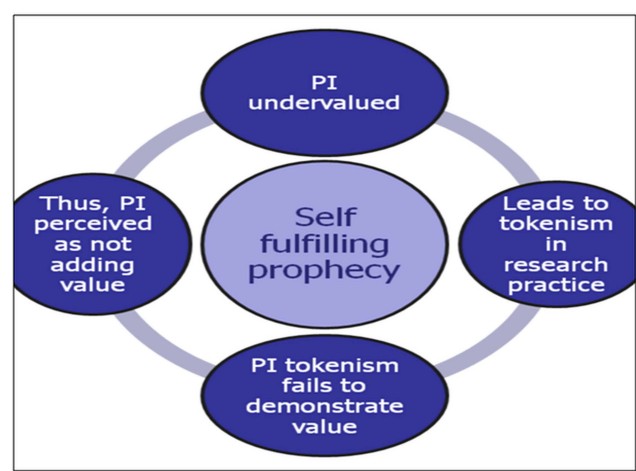

**Figure 2** Public involvement (PI) tokenism: a self-fulfilling prophecy.

non-responding participants at round 1 and round 2 of the survey to stimulate additional responses.[56] Despite a low round 1 response rate, it was encouraging that a large percentage of responders to round 1 subsequently completed round 2. Continued commitment from panellists throughout the Delphi data collection process is required and individual time constraints together with lack of familiarity with the Delphi technique may have prevented some panellists from being able to make such a commitment. However those who did take part were firmly committed to offering us detailed and extremely thoughtful responses to our questions.

A further potential limitation relates to the representativeness of our panel members. First, as described earlier, we opted to use the INVOLVE definition of public,[43] which encompasses patients, potential patients, carers and users of health and social care services. However, we did not ask participants within this stakeholder group to identify themselves more precisely as occupying one or other of these positions. We recognise that there may be clear differences in the views, experiences and resultant contributions of members of the public, depending on their particular position in relation to a research topic; and that this is not captured in our analysis. Identifying any differences in the contributions made to the research process across the different types of 'public' could be a topic for future PI research.

Second, less than 50% of those approached at round 1 participated and this percentage further reduced at round 2. Those opting in to the survey self-selected themselves into a stakeholder group, we therefore hold no information about the groupings of those who opted out; nor do we have information about their other characteristics of interest including, for example, undergoing training in relation to PI. We are therefore unable to comment meaningfully on the representativeness or otherwise of the study population. A final limitation

relates to those opting to take part in the Delphi study as they may represent those with a particularly strong commitment to the PI endeavour, and as such keenly endorse its validity. In light of this our findings may be overly optimistic, which should be considered when interpreting the findings.

## Conclusions and implications for policy and practice

This study is the first, to our knowledge, to present empirical evidence of the opinions of key stakeholders within the health and social care arena about the impacts of PI on the research process; and to identify areas of consensus and conflict around these impacts. We have identified a number of key issues in relation to perceived PI barriers and drivers, perceived impacts of PI and approaches to its evaluation in health and social care research, including:

▶ The potential for tokenism in current PI practice, which requires to be challenged at every stage in the research process;
▶ Agreement that doing PI well can be challenging at the interpersonal and organisational levels;
▶ Difficulties in evaluating the impact of PI as a distinct and individual component of the research process;
▶ Lack of recognition of the value of research team cohesion;
▶ Shortcomings in current provision of appropriate and timely resources, including funding for PI and the provision of PI training and support for members of the public and researchers.

Panellists articulated that the barriers and tensions associated with PI could be addressed by clear guidance on what PI means, together with models of good practice and measurable standards. Several research studies are contributing to this agenda. For example, the wider MRC research within which this Delphi study sits has produced guidance and related resources to support assessment of the impact of PI in research, including draft 'good practice' standards. This is now available online (http://www.piiaf.org.uk). There are also a number of important policy initiatives underway, including work by the Clinical Research Networks in England, to produce standards for PI that will work across the National Institute for Health Research. INVOLVE[43 52] continues to develop guidance and promulgate models of good practice including, most recently a review of work on principles and standards for PI.[57] Concluding that it 'remains unclear how feasible it is to develop standards that are applicable across the range and diversity of involvement activity', INVOLVE has now established an advisory group to explore the feasibility of producing a 'good practice' framework based on principles identified in their review.

Notwithstanding these initiatives it is clear from the findings reported here that individual values and attitudes operating alongside organisational cultures continue to sustain tokenistic practice in PI. While good practice standards have a role to play in shifting these constraints, these will only be effective if they are taken up and promoted by influential international and national research funders who are also committed to sustaining an effective PI infrastructure. This would involve provision of financial support such as for preprotocol work and effective auditing of funded PI activity.

**Author affiliations**
¹Department of Public Health and Policy, University of Liverpool, Liverpool, UK
²Department of Biostatistics, University of Liverpool, Liverpool, UK
³Institute of Health Research, University of Exeter Medical School, Exeter, UK
⁴Division of Health Research, Lancaster University, Lancaster, UK

**Acknowledgements** The authors of this paper are members of the Public Involvement Impact Assessment Framework (PiiAF) Study Group, which involves researchers at the Universities of Lancaster, Exeter and Liverpool. Other members of the PiiAF Study Group who contributed to the Delphi study were: Michelle Collins, Andy Gibson, Elaine Hewis, Jenny Preston, Tim Rawcliffe and Paula Williamson. The authors would also like to acknowledge the valuable input of the members of the Public Advisory Group connected to this study: Bert Green, Faith Harris-Golesworthy, Dina Lew, Irene McGill, Nigel Pyart; and members of the Advisory Network. They are indebted to all those who took part in the Delphi study and wish to thank participants for their time and insight. The PiiAF is accessible via http://www.piiaf.org.uk.

**Contributors** DS was responsible for day-to-day management of the Delphi study, participation in the conduct of the workshops and development of the survey questionnaires, the qualitative data analysis and the drafting of the manuscript; FG reviewed and commented on the survey questionnaires in light of the literature review he conducted as part of the wider MRC Study; JK was responsible for management and analysis of the quantitative data; JP and NB contributed to the conceptual development of the Delphi study and commented on the manuscript; KF, FL and KW commented on the survey documents and the manuscript; AJ had responsibility for the overall conceptual and methodological development of the Delphi study, supervision of DS, and drafting/finalising of the manuscript. JP was also principal investigator for the PiiAF research overall.

**Funding** The study was supported by the Medical Research Council's Methodology Research Programme [G0902155/93948].

**Competing interests** Professor JP was a member of the MRC Methodology Research Programme at the time this grant was awarded, although had no involvement in the funding decision.

**Ethics approval** University of Liverpool Research Ethics Committee.

**Provenance and peer review** Not commissioned; externally peer reviewed.

**Data sharing statement** The authors are exploring the scope for making the anonymised data files available from the corresponding author at the University of Liverpool, which would provide a permanent, citable and open access home for the data set.

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
