## [Reviewer comments · BMJ Open]

Some articles will have been accepted based in part or entirely on reviews undertaken for other BMJ Group journals. These will be reproduced where possible.

ARTICLE DETAILS

TITLE (PROVISIONAL)	Exploring perceived barriers, drivers, impacts and the need for evaluation of public involvement in health and social care research: A modified Delphi Study
AUTHORS	Snape, Darlene; Kirkham, Jamie; Britten, Nicky; Froggatt, Katherine; Gradinger, Felix; Lobban, Fiona; Popay, Jennie; Wyatt, Katrina; Jacoby, Ann

VERSION 1 - REVIEW

REVIEWER	Jonathan Boote University of Sheffield, United Kingdom
REVIEW RETURNED	05-Mar-2014

GENERAL COMMENTS	This is an interesting and thoughtful paper presenting follow-up findings about a Delphi study investigation into perceived barriers, drivers and impacts relating to public involvement in research. I recommend that the journal considers accepting the paper subject to the authors addressing the following minor comments: 1. page 4, I think the PI acronym should be spelled out at first usage in the main body of the paper.2. page 6. I would suggest that the authors include a diagram with a worked example showing what they mean by critical and clear consensus, as these concepts are quite difficult to follow in my view3. page 9, The sub-title "Issues of impacts of PI" is quite clumsily worded, so could this please be rephrased?4. page 11, section 4.1, 2nd paragraph, the authors may wish to reference 2 recent papers reporting evaluations of PPI funding provided by NIHR Research Design Services: (1): Boote J, Twiddy M, Baird W, Birks Y, Clarke C, Beever D (2013). Supporting public involvement in research design and grant development: a case study of a public involvement award scheme managed by a National Institute for Health Research (NIHR) Research Design Service (RDS). Health Expectations. Early View (online version). DOI: 10.1111/hex.12130; (2) Walker, D-M, Pandya-Wood R (2013), Can research development bursaries for patient and public involvement have a positive impact on grant applications? A UK-based, small-scale service evaluation. Health Expectations. Early View (online version). DOI: 10.1111/hex.12127.5. I think the discussion could be improved by including a section on implications for policy and practice. For example, the conclusion to the paper suggests the need for clear guidance on what PI means, together with models of good practice and measurable standards. Could the authors expand on this a little more and suggest who
---

	is/should be responsible for taking this work forward. 6. page 18, Table 1, I would have liked more detail about what form the pilot consultation process took, and the percentage breakdown of panelists' areas of expertise who completed Rounds 1 and 2.
--	---

REVIEWER	Jonathan Tritter Aston University UK
REVIEW RETURNED	21-Mar-2014

GENERAL COMMENTS	This is a really interesting article that explored the views of clinical and non-clinical academics, members of the public, research managers, commissioners and funders. This is an important area. Most research funders require patient and public involvement in the research but this is rarely assessed except at the point of the submission of the proposal. More problematically it is unclear if involvement changes or improves research. This article draws on a Delphi process to identify points of consensus among the diverse sample of stakeholders with an interest in patient and public involvement in research. The definition of patient and public involvement only uses the term 'public' but refers to the INVOLVE definition of 'public' includes patients and potential patients, carers and people who use health and social care services.' I think this is problematic. If people are involved in research because of their experience of services directly as patients or indirectly as carers or family members then their perspective is different from participating as a member of the public. Greater debate on the distinctions between these different roles rather than an assumption that they are synonymous in terms of participation in research would be helpful. I recognise that the research applied a particular definition but I do not think that definition is sufficient to draw distinctions in the different ways non-researchers participate in research. The role that the involved non-researcher plays is important and is likely to vary with different tasks in the research process. There are some important findings. The view of respondents that the integration of involvement makes it difficult to differentiate impact on discrete aspects of the research. Perhaps if involvement does become integrated then the involved person is an integral part of the research it becomes more difficult to assign contribution to a specific member of the research team. Particular research tasks may be associated with an individual but contributions to thinking, analysis and interpretation are far more difficult to disentangle. There was no consensus that involvement produced better quality or more relevant research but that it could increase the impact as those involved could access other dissemination routes and add legitimacy to the presentation of the findings. This is an interesting set of findings. I wonder if part of what is being identified is that involvement changes the process and experience of research but that the identification of evidence that the 'new' process is better than traditional research is hard to specify. Overall this is a clear, well-written article that makes a significant contribution to current and important debates concerning the
--

VERSION 1 – AUTHOR RESPONSE

Reviewer 1:

P.4 – we have spelled out the acronym, PI, as requested in the main body of the paper.

P.6 – we have now added a new Table 2, with illustrative examples of ‘clear’ and ‘critical’ consensus.

P.9 – the sub-title, ‘Issues of impacts of PI’ has been re-worded as requested.

P.11 – we have amended the text to include reference to the articles by Boote et al and Walker et al, as suggested, and thank the reviewer for directing us to these.

P.13 – we have amended the Conclusions section to include discussion of policy and practice implications, as we see them, of our research.

Reviewer 2:

We thank the reviewer for his very supportive statements about our work and agree with his comments about the difficulties in differentiating the impacts of PI from other types of inputs; and that PI may nonetheless add legitimacy to research findings.

We note his comment about the definitional issues around the word, ‘public’ and have added further text in the ‘study limitations’ section, discussing this point and acknowledging that we did not seek to distinguish the different constituent groups, one from another.

Finally, we would like to thank both reviewers for their helpful and very positive comments about the paper. We hope they and you will feel we have responded appropriately and adequately to them; and look forward to hearing your decision in due course.